# Co-Incubation with PPARβ/δ Agonists and Antagonists Modeled Using Computational Chemistry: Effect on LPS Induced Inflammatory Markers in Pulmonary Artery

**DOI:** 10.3390/ijms22063158

**Published:** 2021-03-19

**Authors:** Noelia Perez Diaz, Lisa A. Lione, Victoria Hutter, Louise S. Mackenzie

**Affiliations:** 1School of Life and Medical Sciences, University of Hertfordshire, Hatfield AL10 9AB, UK; n.perez-diaz@herts.ac.uk (N.P.D.); l.lione@herts.ac.uk (L.A.L.); v.hutter@herts.ac.uk (V.H.); 2School of Pharmacy and Biomolecular Sciences, University of Brighton, Brighton BN2 4GJ, UK

**Keywords:** nuclear receptor, gene transcription, inflammation, molecular docking, PPARβ/δ, inflammation, lung, pulmonary artery, GW0742, GSK3787, docking, lipopolysaccharide (LPS)

## Abstract

Peroxisome proliferator activated receptor beta/delta (PPARβ/δ) is a nuclear receptor ubiquitously expressed in cells, whose signaling controls inflammation. There are large discrepancies in understanding the complex role of PPARβ/δ in disease, having both anti- and pro-effects on inflammation. After ligand activation, PPARβ/δ regulates genes by two different mechanisms; induction and transrepression, the effects of which are difficult to differentiate directly. We studied the PPARβ/δ-regulation of lipopolysaccharide (LPS) induced inflammation (indicated by release of nitrite and IL-6) of rat pulmonary artery, using different combinations of agonists (GW0742 or L−165402) and antagonists (GSK3787 or GSK0660). LPS induced release of NO and IL-6 is not significantly reduced by incubation with PPARβ/δ ligands (either agonist or antagonist), however, co-incubation with an agonist and antagonist significantly reduces LPS-induced nitrite production and *Nos2* mRNA expression. In contrast, incubation with LPS and PPARβ/δ agonists leads to a significant increase in *Pdk−4* and *Angptl−4* mRNA expression, which is significantly decreased in the presence of PPARβ/δ antagonists. Docking using computational chemistry methods indicates that PPARβ/δ agonists form polar bonds with His287, His413 and Tyr437, while antagonists are more promiscuous about which amino acids they bind to, although they are very prone to bind Thr252 and Asn307. Dual binding in the PPARβ/δ binding pocket indicates the ligands retain similar binding energies, which suggests that co-incubation with both agonist and antagonist does not prevent the specific binding of each other to the large PPARβ/δ binding pocket. To our knowledge, this is the first time that the possibility of binding two ligands simultaneously into the PPARβ/δ binding pocket has been explored. Agonist binding followed by antagonist simultaneously switches the PPARβ/δ mode of action from induction to transrepression, which is linked with an increase in *Nos2* mRNA expression and nitrite production.

## 1. Introduction

PPARβ/δ are ligand dependent transcription factors that belong to the nuclear receptor family [1]. They are ubiquitously expressed in all cells tested [2] and control key biological functions such as inflammation, metabolism, cell proliferation and migration [3,4,5]. Consequently, agonists and antagonists for PPARβ/δ have been studied as potential therapies for a wide range of diseases and conditions. However, they have failed to lead to a marketed drug which may be linked to a fundamental lack of understanding of the complexity by which PPARβ/δ controls cell function through gene induction and transrepression.

In vivo studies (mice, rats and rhesus monkeys) indicate that PPARβ/δ agonists induce several favorable pharmacological effects: reduced weight gain, increased metabolism in the skeletal muscle and cardiovascular function, suppression of atherogenic inflammation as well as improvement of the blood lipid profile, all of which are common abnormalities in patients with metabolic syndrome [3,6,7]. These encouraging results led to the first clinical trials on humans. Glaxo Smith Kline (GSK) developed the agonist GW501516 (Endurobol), a promising compound that completed proof-of-concept clinical trials successfully for dyslipidaemia [8] and hypocholesteraemia [9]. Further studies revealed a potential link with tumor development [10,11], and any further clinical trial with GW501516 was suspended.

Nevertheless, the interest in PPARβ/δ continued, and in the last few years several compounds targeting PPARβ/δ were developed and entered clinical trials. The Phase II clinical trial on the PPARβ/δ agonist MBX−8025 for treatment of non-alcoholic steatohepatitis and primary sclerosing cholangitis was terminated early when patients developed early signs of liver damage [12]. This leads to questions on how ligands are binding to the receptor to induce different cellular outcomes.

PPARβ/δ can be activated by numerous endogenous ligands such as eicosanoids, fatty acid, metabolites derived from arachidonic acid and linoleic acid [13,14,15] as well as exogenous synthetic ligands like GW0742, L−165041, MBX−8025 and GW501516; whereas PPARβ/δ can also be inhibited by two synthetic antagonists GSK3787 (irreversible) and GSK0660 (competitive). There is a great deal of complexity in the manner by which agonists and antagonists control PPARβ/δ signaling, and the resulting changes in gene expression controls the functional outcome of the cell.

The PPARβ/δ endogenous and exogenous ligands control cellular function through changes in very small concentration range. Added to this, in any cell or tissue, the activity of PPARβ/δ may also depend on its promoter activity and relative expression, as well as presence and activity of co-repressor and co-activator proteins. It has been shown that GW0742 is capable of behaving as an agonist activating the transcription pathway at lower concentrations (nM) and antagonist inhibiting this effect at higher concentrations (μM) [16]. In the same line, a study in a model of systemic inflammation in mice showed that higher doses of GW0742 (0.3 mg/kg) triggered a pro-inflammatory response, whereas a lower concentration (0.03 mg/kg) showed an anti-inflammatory trend, although without a significant difference [17]. It was suggested that the large variation in results may be due to the binding of more than one ligand in the large PPARβ/δ ligand binding domain, which requires further investigation.

After ligand activation, PPARβ/δ regulates genes by two different mechanisms, induction and transrepression. In the induction mode, PPARβ/δ forms a complex with the retinoid X receptor (RXR) and together, as a heterodimer, binds the promoter of the target genes (PPRE). In the absence of ligand, co-repressor proteins and histone deacetylases (HDACs) are bound to the heterodimer which tightens the chromatin and prevent it from binding to the PPRE sites [18]. The presence of ligand induces a conformational change of PPARβ/δ which promotes the binding of co-activators, releases the co-repressor proteins, induces histone acetylation and methylation and finally allows the transcription of the target genes [19,20].

In the transrepression mode PPARβ/δ regulates gene expression in a PPRE-independent manner through the regulation (mostly suppression) of other transcription factors, including nuclear factor-κB (NF-κB) [21], activator protein 1 (AP−1) [22] and B cell lymphoma 6 (Bcl6) [23]. There are great discrepancies in the literature about the effect of the ligand-activation of PPARβ/δ in the cell, and both pro- and anti- effects in inflammation [24,25], cell proliferation [26,27] and migration [28,29] have been reported.

In order to isolate the transrepression mode of PPARβ/δ, a novel approach was taken in this project. Tissues were co-incubated with both agonist; this will alter the conformational shape of PPARβ/δ and bind to co-activators and place the PPARβ/δ predominantly into gene induction mode. In theory, addition of an antagonist will then prevent PPRE binding and induction of genes, revealing the effects of PPARβ/δ transrepression.

Studies have generally focussed on the effects of agonists and antagonists separately, and often results in conflicting theories of how PPARβ/δ controls gene expression. Here we show for the first time that co-incubation of agonists and antagonists to PPARβ/δ leads to a significant decrease in LPS-induced inflammation in rat pulmonary artery compared to single applications of each drug type. The mechanism of action may be explained by the binding studies indicated by docking studies of the agonists and antagonists with PPARβ/δ, which confirms that agonist and antagonist co-binding can occur.

## 2. Results

### 2.1. PPARβ/δ Expression and Basal NO Production Over Time in Pulmonary Artery

Expression of PPARβ/δ was confirmed by ELISA and calculated to 0.04 pg/mL/μg protein. Rat lung pulmonary arteries were incubated with LPS for different periods of time (8 h, 20 h and 24 h) in order to ascertain the minimum time required to significantly increase NO production. LPS induced a significant increase in NO at 24 h in all tissues tested (Figure A1), subsequently the 24 h incubation time period was used.

### 2.2. PPARβ/δ Regulation of LPS-Induced Inflammation

Pulmonary arteries exposed to LPS for 24 h significantly increased production of NO and IL-6, markers of innate inflammation (Figure 1). Incubation of either PPARβ/δ agonists (GW0742 or L−165041), or antagonist (GSK3787 or GSK0660) had no effect on nitrite production (a measure of NO release) (Figure 1A,B) or IL-6 release (Figure 1C,D). In contrast, co-incubation with a mixture of both GW0742 (agonist) plus GSK3787 (irreversible antagonist) (Figure 1A,C) or L−165041 (agonist) plus GSK0660 (competitive antagonist) (Figure 1B,D) led to a significant decrease in NO and IL-6 production.

Production of nitrite is a measure of NO release from cells and tissues; the iNOS specific inhibitor 1400W was used to indicate the proportion of NO originating from iNOS as opposed to eNOS or iNOS. In all experiments, nitrite production following LPS incubation was significantly decreased by incubation with 1400W (Figure 1A,B).

### 2.3. Marker Genes for PPARβ/δ Induction and Transrepression in Pulmonary Artery

In pulmonary artery incubated in LPS, GW0742 significantly increases the transcription of *Pdk−4* mRNA by 7-fold (Figure 2A) and *Angptl−4* mRNA by 3-fold (Figure 2B), which is inhibited by the irreversible antagonist GSK3787. The expression of *Nos2* mRNA, the gene that encodes for iNOS, is significantly increased by LPS, and its expression is significantly inhibited by co-incubation with GW0742 and GSK3787 (Figure 2C).

### 2.4. Computational Chemistry: PPARβ/δ Docking Analysis

#### 2.4.1. Docking of One PPARβ/δ Ligand

The PPARβ/δ-LBD crystal structure 3TKM has an X-ray resolution of 1.95 Å and was co-crystallized with GW0742, the same agonist that was used during the development of this project, therefore this structure was chosen for our docking experiments.

The two PPARβ/δ agonists GW0742 and L−165041 as well as the two antagonists GSK3787 and GSK0660 were docked into the crystal structure of the LBD of PPARβ/δ. The best eight hits were analyzed by Pymol to identify the residues that form polar interactions with each of the different poses of the ligands (Table 1).

#### 2.4.2. Docking of GW0742

The most stable orientation of GW0742 within the PPARβ/δ binding pocket predicted by Autodock Vina (green) was compared to the real GW0742 present in the crystal structure (pink) (Figure 3A). The more detailed image (Figure 3B) clearly shows that the residues that form polar interactions with GW0742 are His247, His413 and Tyr437 (Pymol), whereas the 2D image created by Ligplot+ showing how the head of GW0742 forms the polar bindings and the tail is surrounded by the hydrophobic amino acids (Figure 3C).

#### 2.4.3. Docking of GSK3787

GSK3787 binds in a slightly different place than GW0742, although there is some overlapping of the binding sites (Figure 4A). Also, the amino acids involved in the polar interaction of GSK3787 predicted by Pymol, Thr252 and Asn307, are different to those of the agonists (Figure 4B) as well as the residues that interact with the hydrophobic tail of GSK3787 (Figure 4C).

#### 2.4.4. Docking of L−165042

The most stable L−165041 orientation predicted by Autodock Vina binds in the same physical place as GW0742 (Figure 5A) and the same three amino acids (His287, His413, Tyr437) form polar interactions with the head of L−165041 (Figure 5B). The tail of L−165041 also forms hydrophobic interactions with a number of residues in common with GW0742, such as Val245, Arg248, Cys249, Thr253, Phe291, Leu294, Val305, Val312, Met417, Leu433 (Figure 5C).

#### 2.4.5. Docking of GSK0660

GSK0660 binds very close but not in the same binding site as GW0742 (Figure 6A). The amino acids involved in the polar bindings with GSK0660, Thr252, Asn307, Arg248 and Ala306, are again different to those for the agonists, although two of them are common with GSK3787 (Figure 6B). Ligplot+ predicts slightly different polar binding profile (Figure 6C), probably because these two software’s use different algorithms for binding prediction, although still show hydrophobic interactions common with GSK3787, such as Trp228, Val305 and Ala306.

#### 2.4.6. Docking of Two PPARβ/δ Ligands Simultaneously

In order to investigate the docking of two ligands simultaneously, the first ligand was bound in the most stable orientation first. The best hit from previous docking was assigned Ligand 1, and then a second molecule was docked, assigned Ligand 2. The aim was to mimic the conditions of the experiments performed in this study and predict what could have happened at the molecular level. When the tissues were treated with only one ligand there is only one option for two ligands to bind, but when two different ligands are present at the same time either of them can bind first into the binding pocket. All these ligand-binding possibilities were considered and summarized in Table 2. A further analysis on Pymol was completed for each option.

#### 2.4.7. Analysis of GW0742 and GSK3787 Docked into GW0742-Bound PPARβ/δ

Once GW0742 is bound in the most stable orientation within the PPARβ/δ-LBD, GW0742 and GSK3787 can still bind at favorable energies (−8.5 kcal/mol and −7.7 kcal/mol respectively), although at very different binding sites to the most stable one and forming polar interactions with different residues, as shown in Figure 7.

#### 2.4.8. Analysis of GW0742 and GSK3787 Docked into GSK3787-Bound PPARβ/δ

GW0742 and GSK3787 can also bind into the binding pocket after GSK3787 at favorable energies (−8.1 kcal/mol and −7.4 kcal/mol respectively). The binding site is also different to the most stable ones (Figure 8), but interestingly the binding site is also different to the previous one, when GW0742 is bound first into the binding pocket instead of GSK3787 (Table 2).

#### 2.4.9. Analysis of L−165041 and GSK0660 Docked into L−165041-Bound PPARβ/δ

When L−165041 is bound to the ligand binding pocket first, a second molecule of L−165041 or GSK0660 can bind with favorable energies (−8.3 kcal/mol and −6.5 kcal/mol respectably) and again, forming polar interactions with different residues. The most interesting finding is that GSK0660, although still in the PPARβ/δ-LBD, has the potential to bind outside the binding pocket (Table 2).

#### 2.4.10. Analysis of L−165041 and GSK0660 Docked into GSK0660-Bound PPARβ/δ

L−165041 and GSK0660 can also bind within the binding pocket at favorable energies (−8.1 kcal/mol and −8.9 kcal/mol respectively) after GSK0660 is bound in the most stable orientation (Table 2), but again the binding site is different to previously when L−165041 was bound first.

## 3. Discussion

The study confirms previous studies that PPARβ/δ is expressed in pulmonary arteries [30]. Here we show for the first time that dual incubation of PPARβ/δ ligands (agonist plus antagonist) decreases LPS-induced NO and IL-6 release from pulmonary arteries compared to single incubation with either agonist or antagonist alone. This result was repeatable with two combinations of agonists and antagonists. Docking studies using computational chemistry methods indicates that multiple binding of both types of ligand is possible and retain similar binding profiles as when binding alone.

Other in vivo studies in whole mouse have indicated that GW0742 treatment attenuates inflammation, which is significantly reduced by GSK0660 [31,32,33]. These in vivo studies differ from our ex vivo studies in isolated tissues, as they are not affected by the immune system infiltration of the alveolar cavity. In our ex-vivo pulmonary artery LPS model, PPARβ/δ was shown to regulate NO and IL-6 release only under certain conditions, which required further investigation.

Marker genes were selected to represent the induction mode of action of PPARβ/δ. Khozoie [34] and Adhikary [35] performed a genome wide analysis of genes regulated by PPARβ/δ in mouse keratinocytes and human myofibroblasts respectively. Khozoie [34] cross-linked the two lists of genes regulated by PPARβ/δ and created a new list of 103 genes regulated by PPARβ/δ in both human and mouse models. There is a high possibility that these genes are regulated by PPARβ/δ in rats as well, therefore this list was used to select the PPARβ/δ induction marker genes *Angptl−4* and *Pdk−4*. Here in this study, qRT-PCR of pulmonary artery showed that the activation of PPARβ/δ with GW0742 increases the transcription of *Pdk−4* and *Angptl−4* mRNA indicating that agonist activation of PPARβ/δ triggers induction, which was inhibited by co-incubation with GSK3787.

It has been shown in several studies that *Nos2* mRNA is regulated by PPARβ/δ [33,36,37], although it has not been described whether this regulation is via induction or transrepression. This study indicates that the significant increase in LPS induced *Nos2* mRNA expression is not altered by the presence of either GW0742 or GSK3787 which indicates that direct induction of genes is not the predominant mode of control PPARβ/δ has on *Nos2* mRNA expression. However, *Nos2* mRNA expression is significantly reduced when co-incubated with both agonist and antagonist. With the activation of the receptor followed by inactivation by antagonist, any reduction in expression and activity of iNOS must be due to transrepression of PPARβ/δ on other nuclear receptors.

It has been suggested that the large ligand binding pocket of PPARβ/δ can accommodate more than one ligand, resulting in unusual PPAR:ligand stoichiometries that could trigger inactive receptor conformations [16], a possibility that we further investigated using in silico methods. Firstly, the two PPARβ/δ agonists (GW0742 and L−165042) and two antagonists (GSK3787 and GSK0660) were docked into the PPARβ/δ binding site. It was found that the agonists and antagonists have a different binding profile within the binding pocket: the same three amino acids His287, His413 and Tyr437 form polar interactions with the two agonists tested, but they do not bind the antagonists. Whereas the amino acids Thr252 and Asn307 are more prone to bind the antagonists GSK3787 and GSK0660.

This finding agrees with previous results were GW0742 was docked to another PPARβ/δ crystal structure (PDB: 3GZ9) using another docking software (Glide), and the same three amino acids bound to GW0742 [38]. Furthermore, several studies co-crystallized PPARβ/δ with different agonists both synthetic, such as iloprost [39], the fibrate GW2433 [40], or GW501516 [41] and natural PPARβ/δ agonists such as with eicosapentaenoic acid (EPA) [40], and in all cases the agonists showed polar bindings with the same three amino acids His287, His413 and Tyr437.

It is worth mentioning another study where the authors selected 5 compounds that potentially bound PPARβ/δ and performed a luciferase transactivation assay to biologically test if these compounds activate PPARβ/δ. They further analyzed two of them by docking and molecular dynamics (MD) simulation, one compound that activated PPARβ/δ (Compound **1**) and another one that did not activate PPARβ/δ (Compound **2**). The docking and MD simulation results for the Compound **1** showed an interaction with His287, His413 and Tyr437, and the results for Compound **2** showed an interaction with Thr252 [42]. This suggests the possibility that the different binding profile between agonists and antagonists can provoke a different 3D conformational change that might explain why PPARβ/δ binds to co-repressors instead of co-activators and vice versa.

Our findings clearly indicate that a ligand that shows a high binding affinity and is predicted to form polar bonds with His287, His413 and Tyr437 will most likely behave as agonist. On the other hand, if one ligand shows high binding affinity but it is predicted to bind other residues such as Thr252 and Asn307 it is more likely that it will behave as antagonist.

Co-incubation of pulmonary artery with two types of ligands led to unexpected results, raising the question on whether the agonist or antagonist retained the potential to bind PPARβ/δ in the expected manner. Incubation with only one agonist GW0742 allows two possibilities: the binding of one or two molecules into the ligand-binding domain (LBD). If a second molecule of GW0742 binds to PPARβ/δ, this molecule is predicted to bind not too far from the most stable binding site and with the same binding affinity and residue interaction (Arg258) than the 8th best position predicted for the first molecule. Similarly, when the irreversible antagonist GSK3787 is present, a second molecule of GSK3787 is predicted to bind also not far away from its most stable binding site and with favorable binding affinity. In addition, it is predicted that GSK3787 will still form polar bonds with Thr252, an amino acid that is predicted to bind GSK3787 in five out of eight most stable poses predicted by docking.

When investigating dual occupancy of the PPARβ/δ LBD with both agonist (GW0742) and antagonist (GSK3787) all the options mentioned above still apply but two more options are available: GW0742 binds first and GSK3787 after or GSK3787 binds first and GW0742 after. When GW0742 binds first, GSK3787 can still bind very close to its most stable binding site with a very favorable binding affinity, and what is more, still binds the residue Thr252. If GSK3787 binds first, GW0742 is predicted to bind very far away from the most stable binding site, at the entrance of the binding pocket, and as a consequence it will have a very different binding profile forming polar bonds with Trp228 and Lys229, two residues that did not show any interaction with ligands before. This suggests that if the PPARβ/δ receptor is inhibited by GSK3787, the agonist cannot reverse this inhibition. Similar analysis can be done with the other pair of agonist and antagonist L−165041 and GSK0660.

Taking into account the docking scores and molecular poses of the ligands, all possibilities described above have very favorable energies for it to happen. That opens a whole new scenario of possibilities that could dramatically change the 3D conformation of PPARβ/δ in ways that have not been thought of before, resulting in the binding of different co-regulators, which ultimately could change the PPARβ/δ response from induction to transrepression or vice versa.

To our knowledge, this is the first time that the possibility of binding two ligands simultaneously into the PPARβ/δ binding pocket has been explored. The results suggest that this possibility is very likely to happen with very favorable affinity energies, and it is worth considering when designing and interpreting experiments where PPARβ/δ is ligand-activated and high concentrations of ligands are used. In regards to the conditions set in our study, docking indicates that the co-incubation with both agonist and antagonist does not prevent the specific binding of each other to the large PPARβ/δ binding pocket.

## 4. Conclusions

In summary, this is a multidisciplinary approach of the study of PPARβ/δ that provides novel information about its functioning at molecular level. In the model used here, the simultaneous co-incubation of pulmonary with both agonist and antagonist potentially opens a window to understand the alternative transrepression PPARβ/δ mode of action as compared to the induction mode of action. Here we show for the first time that there is a characteristic PPARβ/δ-ligand binding profile for agonists and antagonists even in combination; PPARβ/δ agonists form polar bonds with His287, His413 and Tyr437, while antagonists are more promiscuous about which amino acids they bind to, although they are very prone to bind Thr252 and Asn307. In dual predictive docking studies, all the options studied seem feasible with favorable binding energies, suggesting the need for caution when designing and interpreting the results of experiments using PPARβ/δ ligands.

## 5. Materials and Methods

### 5.1. Reagents

The PPARβ/δ ligands GW0742, GSK3787, L−165042 and GSK0660 as well as 1400W, LPS O55:B5, sulfanilamide and naphthylethylenediamine dihydrochloride were purchased from Sigma (Gillingham, Dorset, UK). Sodium nitrate, orthophosphoric acid and DMSO were purchased from Fisher Scientific (Loughborough, UK). Primers from Applied Biosystem (Foster City, CA, USA): β-actin (Rn00667869_m1), Pdk−4 (Rn00585577_m1), Angptl−4 (Rn015228817_m1), Nos2 (Rn00561646_m1).

### 5.2. Animals

Male Wistar rats (350–450 g) were sourced from Charles River (Harlow, UK) and housed in pairs in standard cages (Tecniplast 2000P) with sawdust (Datesand grade 7 substrate) and shredded paper wool bedding with water and food (5LF2 10% protein LabDiet) in the Biological Services Unit at the University of Hertfordshire. The housing environment was maintained at a constant temperature of 22 ± 2 °C, under a 12 h light/dark cycle (lights on: 07:00 to 19:00 h). All testing was conducted under the light phase of the animals’ light/dark cycle, and care was taken to randomize treatment sequences to control for possible order effects.

All experiments involving protected animals were carried out in accordance with the University of Hertfordshire animal welfare ethical guidelines and the Animals (Scientific Procedures) Act 1986 and European directive 2010/63/EU. Rats were euthanized according to schedule 1 procedure by CO_2_ asphyxiation followed by cervical dislocation. Lungs were removed and immediately placed in physiological saline solution (PSS) buffer (118 mM NaCl, 4.7 mM KCl, 2.5 mM CaCl_2_, 1.17 mM MgSO_4_, 1 mM KH_2_PO_4_, 5.5 mM, glucose, 25 mM NaHCO_3_ and 0.03 mM Na_2_EDTA). Following dissection, tissues were incubated in 1% *v*/*v* penicillin/streptomycin in DMEM (no serum) under 5% *v*/*v* CO_2_ at 37 °C in the required treatment (detailed in Table 3) for up to 24 h. Whole arteries can be incubated in serum free media for 4 days without significant loss of contraction and differentiation status or death [43]. After incubation, the culture medium was removed and stored at −20 °C until Greiss assay or IL−6 ELISA analysis. The tissues were stored at −80 °C until needed for mRNA extraction for qRT-PCR.

### 5.3. Quantification of PPARβ/δ Expression in Lung Tissues by Enzyme-Linked Immunosorbent Assay (ELISA)

The expression of PPARβ/δ in pulmonary artery from naïve rats was measured using rat PPARβ/δ ELISA kit (Abbkine; Wuhan, China) according to the manufacturer’s instructions. Briefly, the dissected tissues were homogenized with liquid N_2_ using a mortar and a pestle, and the proteins were extracted in ice-cold phosphate buffered saline (PBS) with proteinase inhibitor cocktail in a ratio 9 mL PBS (137 mM NaCl, 2.7 mM KCl, 10 mM Na_2_HPO_4_, 1.8 mM KH_2_PO_4_, pH 7.4) per g tissue. The samples were then sonicated 3 × 30 s and centrifuged for 5 min at 5000 g. The supernatant was collected, and the protein concentration was quantified by BCA assay. Then, 50 µL of standards and samples were added in duplicate to the plate containing pre-coated anti-PPARβ/δ and incubated 45 min at 37 °C. The wells were washed and incubated with 50 µL of the horseradish peroxidase (HRP)-conjugated detection antibody for another 30 min at 37 °C. The wells were washed again and incubated in the dark with the chromogen solution for 15 min at 37 °C. Finally, the reaction was stopped by adding 50 µL of Stop solution and the plate was immediately measured in a microplate reader. The concentrations of PPARβ/δ were determined by comparison of OD_450_ to a standard curve (0–8 ng/mL).

### 5.4. Quantification of Nitric Oxide Released by Lung Tissues by the Griess Assay

An aliquot of the culture medium (50 µL) thawed on ice was mixed with an equal volume of Griess reagent (mixture of equal volumes Griess reagent 1 and Griess reagent 2 containing sulfanilamide 1% *w*/*v* + orthophosphoric acid 5% *v*/*v* and naphthylethylenediamine dihydrochloride 0.5% w/v respectively). The concentration was determined by comparison of the OD_540_ to a standard curve of solutions of sodium nitrite (0–1 mM).

### 5.5. Quantification of IL−6 Released by Pulmonary Artery by ELISA

The release of IL−6 by pulmonary arteries to the culture medium was measured using Rat IL−6 DuoSet ELISA kit (R&D Systems, Minneapolis, MN, USA) according to the manufacturer’s instructions. Briefly, all samples and standards were conducted in duplicate to a microtiter plate containing the capture antibody. After two hours of incubation the wells were washed and incubated with the detection antibody for another two hours. The wells were washed again and incubated in the dark with Streptavidin-HRP for 20 min. The wells were washed once more and incubated in the dark with substrate solution. After 20 min the stop solution was added and immediately measured in a microplate reader. The readings at 540 nm were subtracted from the readings at 450 nm and the concentrations of samples were determined by comparison with the standard curve (0–8 ng/mL).

### 5.6. Quantitative Real Time-Polymerase Chain Reaction (qRT-PCR)

Total RNA was extracted from tissues using RNeasy Fibrous Tissue Mini Kit (Qiagen, Hilden, Germany). The tissues were first pulverized in a pestle with liquid N_2_ and the RNA was then extracted following the manufacturer’s instruction and stored at −80 °C until use. The quality and concentration of the RNA was measured using Nanodrop (SimpliNano, GE Healthcare Life Science; Chalfont St Giles, Buckinghamshire, UK) at a wavelength of 260 nm. Further to that, the degradation of RNA was checked in a 1% *w*/*v* agarose gel, and the DNA contamination of the RNA was checked by PCR. To do that, genomic DNA was also extracted from the tissues using PureLink Genomic DNA mini kit from Invitrogen following the manufacturer’s instructions and used as a positive control of the PCR, and two primers for the housekeeping gene *β-actin* were designed (forward CTGGTCGTACCACTGGCATT, reverse AATGCCTGGGTACATGGTGG). The termociclator Mastercycler Nexus gradient (Eppendorf, Hamburg, Germany) was set with the following PCR protocol: 95 °C for 10 min, 40 cycles of 95 °C for 30 s, 56 °C for 30 s and 72 °C for 30 s, 72 °C for 10 min and hold at 4 °C.

cDNA was obtained by reverse transcription (RT) using iScript cDNA synthesis kit (Bio-Rad) following the manufacturer’s instructions. The RT was performed using a thermociclator Stratagene Mx3005P (Agilent Technologies, Santa Clara, CA, USA) with the following steps: 5 min at 25 °C, 20 min at 46 °C, 5 min at 95 °C, and hold at 4 °C.

qRT-PCR was performed to analyze mRNA expression using a Taqman System. Briefly, 10 µL of reaction mix containing the primers and cDNA was incubated in a 96 well-plate following the cycle conditions: 95 °C for 10 min, 40 cycles of 95 °C for 15 s and 60 °C for 1 min.

### 5.7. Docking

The ability of drugs to bind into protein active sites was investigated using AutoDock/Vina with Pymol and Ligplot+ as a graphical user interface. For the docking simulations, the PPARβ/δ crystal structure 3TKM was selected for having one of the highest resolutions (1.95 Å). The PDB file was downloaded from the Protein Data Bank. Water molecules, ligands and other hetero atoms were removed from the protein structure, and the addition of hydrogen atoms to the protein was performed using AutoDock Tools version 1.5.6. The grid was set manually to cover the active site. The file was saved as a pdbqt file.

The ligand molecule structures were drawn in ChemSketch, the energy was minimized and saved in PDB format, and converted into a pdbqt file with AutoDock Tools version 1.5.6 (ADT/AutoDockTools—AutoDock (scripps.edu) (accessed on 23 February 2021)).

Molecular docking was performed with the software AutoDock Vina and all parameters set as default. Results with minor calculated free energy variations were analyzed using Pymol version 1.7.4 and LigPlot+ version1.4.5 softwares.

For the docking of two molecules, the 3TKM PDB file without hetero atoms was combined with the best docking result of each ligand in one single PDB file, one PDB file per ligand. These files were opened in Autodock Tools, H_2_ were added, the grid was set manually and saved in a new pdbqt file. This file was used for the docking with the second molecule.

### 5.8. Statistical Analysis

Statistical comparisons were performed on GraphPad Prism 5.0 software using one-way ANOVA with Bonferroni’s post hoc analysis for NO and IL−6 detection assays, and for qRT-PCR. The values are expressed as observed mean ± SEM. Data was normalized previously to the statistical analysis. In short, data from NO and IL−6 detection assays was normalized against the group treatment LPS and expressed as % change. The relative quantification of genes analyzed by qRT-PCR was calculated with the comparative CtΔΔ method, β-actin was used as endogenous control and data was normalized against the control group as a fold change.

Values of *p* < 0.05 were considered statistically significant. When the level of probability (*p*) is less than 0.05 (*), less than 0.01 (**) or less than 0.001 (***), the effect of the difference was regarded as significance.

## Figures and Tables

**Figure 1 ijms-22-03158-f001:**
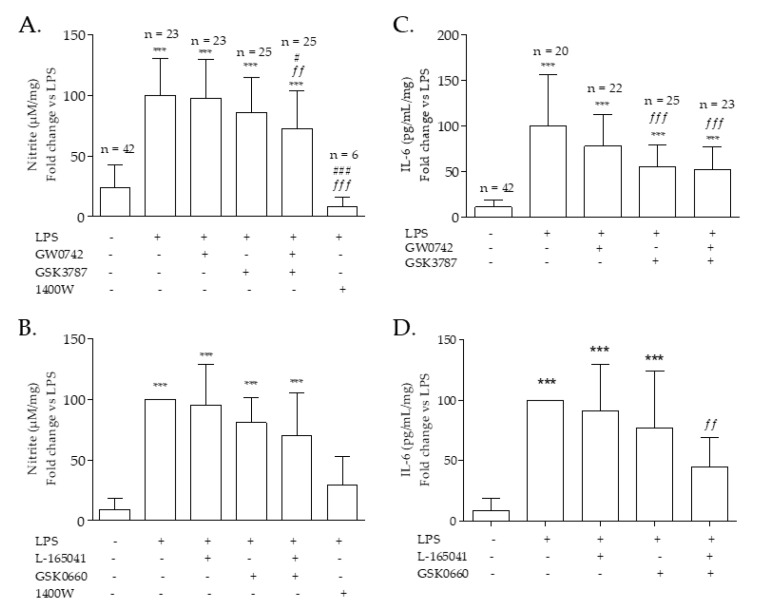
Nitrite (a marker for NO release) and IL−6 production by pulmonary artery. Rat pulmonary artery rings were treated with two combinations of PPARβ/δ agonist-antagonist and iNOS inhibitor 1400W: (**A**–**C**) 100 nM GW0742—1 µM GSK3787—100 µM 1400W (**B**–**D**) 1 µM L−165041—1 µM GSK0660—100 µM 1400W. NO and IL−6 production was measured after 24 h and normalized with LPS: (**A**) the average of NO production in lipopolysaccharide (LPS) incubated tissues was used for the normalization of the data and the number of samples per treatment is written at the top of the bar; (**B**) each experiment was normalized with its own LPS treatment (*n* = 9); (**C**) the average of IL−6 production with LPS treatment was used for the normalization of the data and the number of samples per treatment is written at the top of the bar; (**D**) each experiment was normalized with its own LPS treatment (*n* = 9). Data passed the D’Agostino–Pearson normality test; significant difference between treatments was analyzed by one-way ANOVA followed by Bonferroni post-hoc test and the data are presented as mean ± SD. *** = *p* < 0.001 compared with vehicle; *ff* = *p* < 0.01, *fff* = *p* < 0.001 compared with LPS; # = *p* < 0.05, ### = *p* < 0.001 compared with LPS + GW0742.

**Figure 2 ijms-22-03158-f002:**
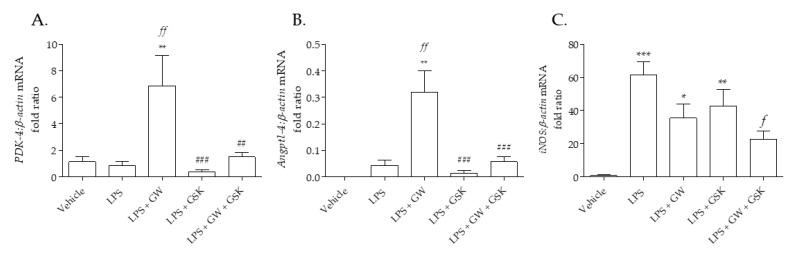
(**A**) *PDK−4*, (**B**) *AngPtl4* and (**C**) *iNOS* mRNA expression in pulmonary arteries following incubation with LPS and PPARβ/δ ligands. The expression of different mRNA was measured following 24 h incubation with treatments: vehicle (0.01% DMSO); 1 µg/mL LPS; 1 µg/mL LPS + 100 nM GW0742; 1 µg/mL LPS + 1 µM GSK3787; and 1 µg/mL LPS + 100 nM GW0742 + 1 µM GSK3787 (*n* = 4–5). Relative quantitation was calculated with the comparative CtΔΔ method and normalized against β-actin as an endogenous control. The data are presented as mean ± standard deviation; the data was not normally distributed (D’Agostino–Pearson normality test). Significant difference by the Krustal–Wallis test with Dunns post hoc test is indicated by * = *p* < 0.05, ** = *p* < 0.01 and *** = *p* < 0.001 compared with Vehicle; *f* = *p* < 0.05, *ff* = *p* < 0.01 compared to LPS; ## = *p* < 0.01 and ### = *p* < 0.001 compared to LPS + GW0742.

**Figure 3 ijms-22-03158-f003:**
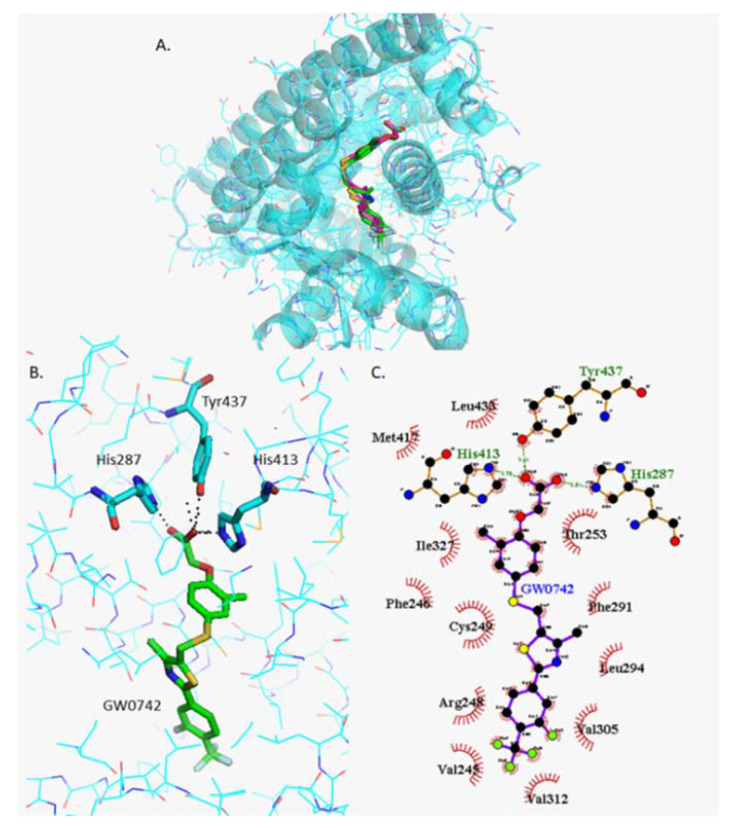
Analysis of GW0742 docked into PPARβ/δ (PBD:3TKM). (**A**) Representation of the most stable GW0742 docking conformation (green) compared to the GW0742 of the crystal structure (pink). (**B**) 3D detail of the amino acids forming polar bindings with GW0742 calculated by Pymol. Colour coding of atoms: red O, blue N, mustard S, white F, pink C of GW0742 from the crystal structure, green C of GW0742 docked into the crystal structure, cyan C from PPARβ/δ. (**C**) Schematic 2D representation of the interaction between PPARβ/δ LBD and GW0742 calculated using Ligplot+. The green dashed lines indicate polar interactions and the red spoked arcs indicate hydrophobic interactions. Colour coding of atoms: red O, blue N, yellow S, green F, black C.

**Figure 4 ijms-22-03158-f004:**
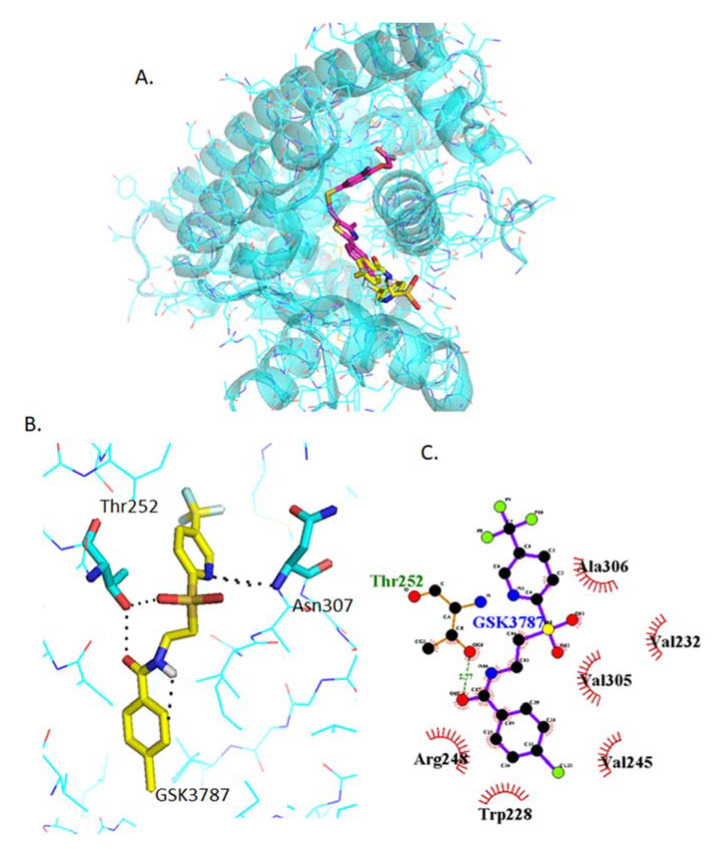
Analysis of GSK3787 docked into PPARβ/δ (PBD:3TKM). (**A**) Representation of the most stable GSK3787 docking conformation (yellow) compared to the GW0742 of the crystal structure (pink). (**B**) 3D detail of the amino acids forming polar bindings with GSK3787 calculated by Pymol. Color coding of atoms: red O, blue N, mustard S, white F, pink C of GW0742 from the crystal structure, yellow C of GSK3787 docked into the crystal structure, cyan C from PPARβ/δ. (**C**) Schematic 2D representation of the interaction between PPARβ/δ LBD and GSK3787 calculated using Ligplot+. The green dashed lines indicate polar interactions and the red spoked arcs indicate hydrophobic interactions. Colour coding of atoms: red O, blue N, yellow S, green F, black C.

**Figure 5 ijms-22-03158-f005:**
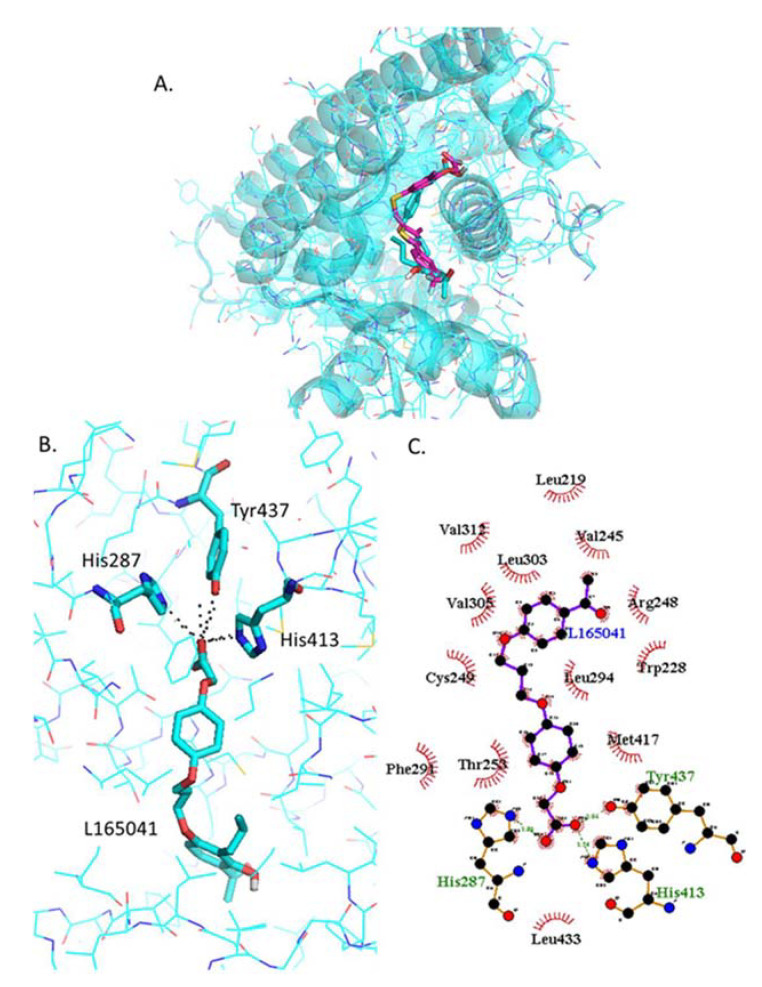
Analysis of L−165042 docked into PPARβ/δ (PBD:3TKM). (**A**) Representation of the most stable L−165041 docking conformation (cyan sticks) compared to the GW0742 of the crystal structure (pink). (**B**) 3D detail of the amino acids forming polar bindings with L−165041 calculated by Pymol. Color coding of atoms: red O, Blue N, mustard S, white F, pink C of GW0742 from the crystal structure, cyan sticks C of L−165041 docked into the crystal structure, cyan lines C from PPARβ/δ. (**C**) Schematic 2D representation of the interaction between PPARβ/δ LBD and -L−165041 calculated using Ligplot+. The green dashed lines indicate polar interactions and the red spoked arcs indicate hydrophobic interactions. Colour coding of atoms: red O, blue N, yellow S, green F, black C.

**Figure 6 ijms-22-03158-f006:**
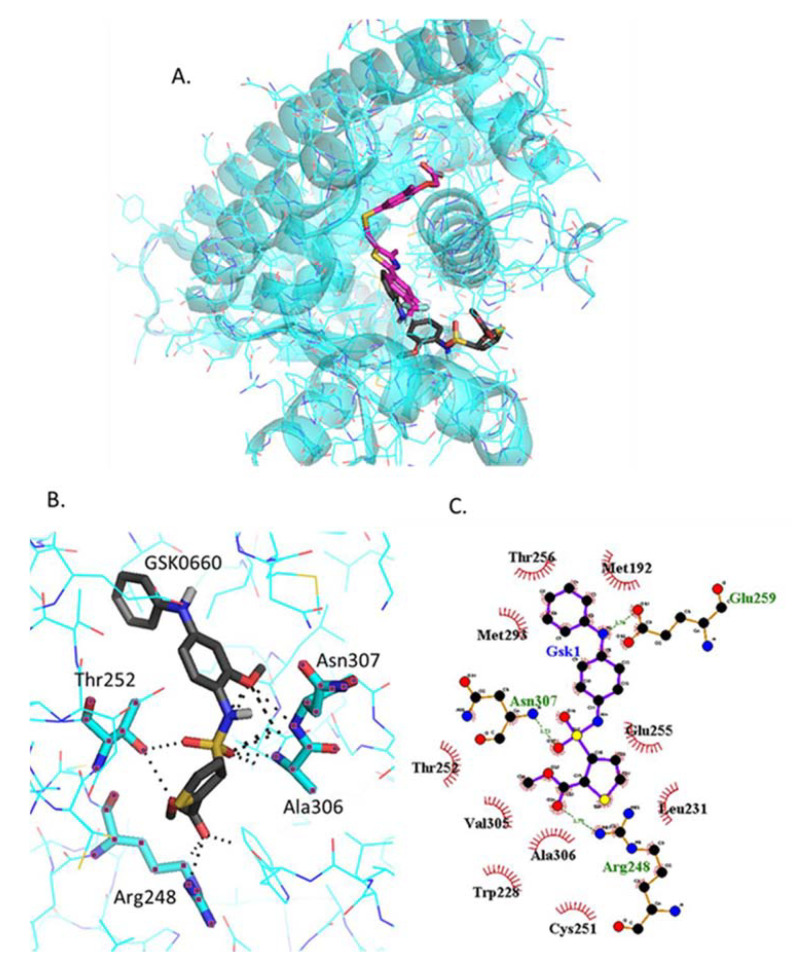
Analysis of GSK0660 docked into PPARβ/δ (PBD:3TKM). (**A**) Representation of the most stable GSK0660 docking conformation (grey) compared to the GW0742 of the crystal structure (pink). (**B**) 3D detail of the amino acids forming polar bindings with GSK0660 calculated by Pymol. Color coding of atoms: red O, Blue N, mustard S, white F, pink C of GW0742 from the crystal structure, grey C of GSK0660 docked into the crystal structure, cyan C from PPARβ/δ. (**C**) Schematic 2D representation of the interaction between PPARβ/δ LBD and GSK0660 calculated using Ligplot+. The green dashed lines indicate polar interactions and the red spoked arcs indicate hydrophobic interactions. Colour coding of atoms: red O, blue N, yellow S, green F, black C.

**Figure 7 ijms-22-03158-f007:**
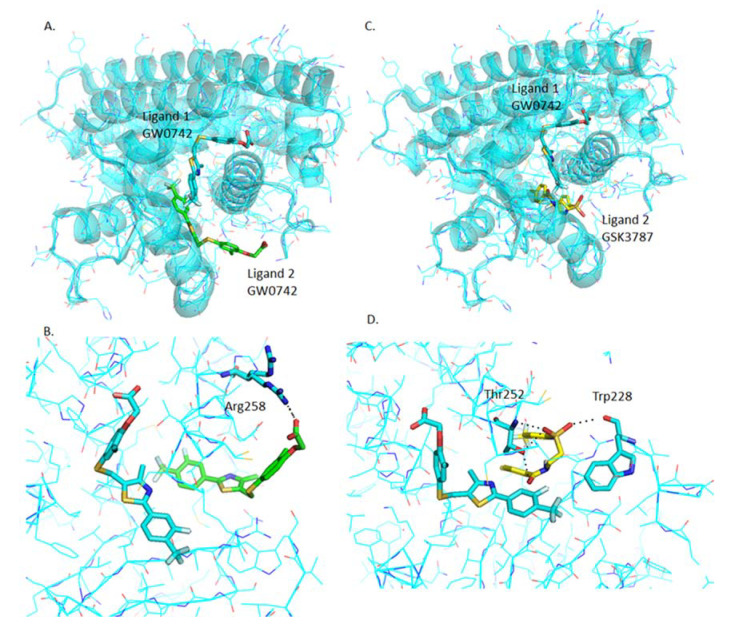
Analysis of GW0742 and GSK3787 docked into PPARβ/δ + GW0742. A second molecule of GW0742 (**A**,**B**) or GSK3787 (**C**,**D**) was docked into the LBD of PPARβ/δ containing one molecule of GW0742. (**A**) Representation of how two GW0742 molecules bind into the PPARβ/δ binding pocket at same time. (**B**) Detail of the amino acids interacting with the second molecule of GW0742. (**C**) Representation of how one molecule of GW0742 first and then one molecule of GSK3787 bind into the PPARβ/δ binding pocket at same time. (**D**) Detail of the amino acids interacting with the second molecule of GSK3787. Colour coding of atoms: red O, blue N, mustard S, white F, cyan C, PPARβ/δ and GW0742 that binds first within the binding pocket, green C of GW0742 that binds second into the binding pocket, yellow C of GSK3787 that binds second into the binding pocket.

**Figure 8 ijms-22-03158-f008:**
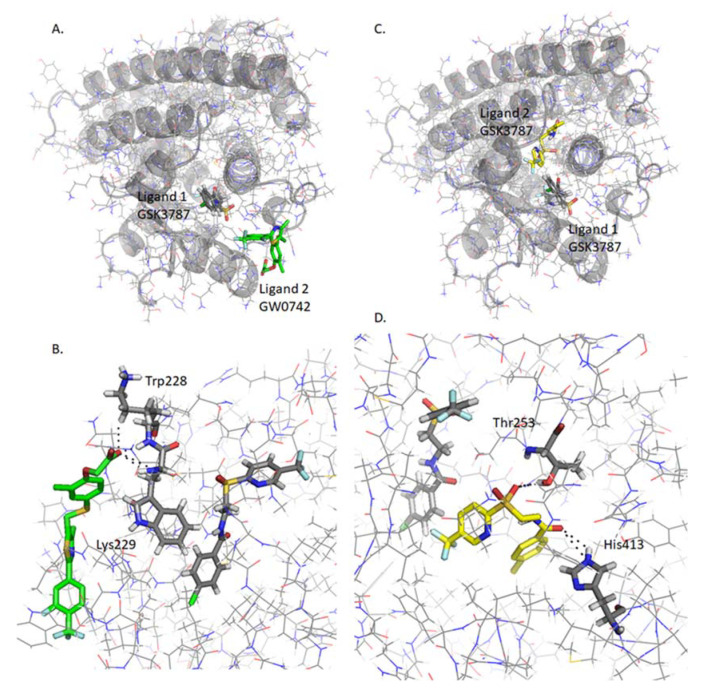
Analysis of GW0742 and GSK3787 docked into PPARβ/δ + GSK3787. A second molecule of GW0742 (**A**,**B**) or GSK3787 (**C**,**D**) was docked into the LBD of PPARβ/δ containing one molecule of GSK3787. (**A**) Representation of how one molecule of GSK3787 first and then one molecule of GW0742 bind into the PPARβ/δ binding pocket at same time. (**B**) Detail of the amino acids interacting with the second molecule of GW0742. (**C**) Representation of how two molecules of GSK3787 bind into the PPARβ/δ binding pocket at same time. (**D**) Detail of the amino acids interacting with the second molecule of GSK3787. Colour coding of atoms: red O, blue N, mustard S, white F, grey C PPARβ/δ and GSK3787 that binds first within the binding pocket, green C of GW0742 that binds second into the binding pocket, yellow C of GSK3787 that binds second into the binding pocket.

**Table 1 ijms-22-03158-t001:** Best eight docking hits of four ligands into PPARβ/δ (PBD: 3TKM).

Best Fit	Agonists				Antagonists			
	GW0742		L-165041		GSK3787		GSK0660	
	Affinity (Kcal/mol)	Aa with Polar Interactions	Affinity (Kcal/mol)	Aa with Polar Interactions	Affinity (Kcal/mol)	Aa with Polar Interactions	Affinity (Kcal/mol)	Aa with Polar Interactions
**1**	−11.1	His287 His413 Tyr437	−8.7	His287 His413 Tyr437	−9.1	Thr252 Asn307	−8.6	Arg248 Thr252 Ala306 Asn307
**2**	−10.8	Thr253 His287 His413 Tyr437	−8	Met192 Thr252 Thr256 Ile290 Ala306	−8.9	Thr252	−8.3	Arg248 Thr252 Ala306
**3**	−9.9	Thr253 His413	−7.6	Thr252 Arg258 Glu259	−8.6	Thr252	−8.1	Thr256 Asn307
**4**	−9.6	Thr256	−7.5	Thr252 Thr253 Ala306	−8.5	Thr256	−8.1	Asn307
**5**	−9.3	No bonds	−7.5	Trp228 Thr252 Thr256 Ile290	−8.4	No bonds	−7.9	Thr256 Ala306 Asn307
**6**	−8.8	Thr253	−7.1	Thr252 Thr256 Ala306	−8.3	Thr252 Thr253	−7.6	Asn307
**7**	−8.7	Thr252	−6.9	Tyr284 Arg361	−8.3	Thr252	−7.5	Ala306 Asn307
**8**	−8.4	Arg258	−6.8	Glu255 Asn307	−8.2	Met192	−7.4	Thr256 Asn307

**Table 2 ijms-22-03158-t002:** Docking prediction of binding affinities and amino acids forming polar interactions with the PPARβ/δ ligands bound into the LBD.

Ligand 1			Ligand 2		
Molecule	Affinity (Kcal/mol)	Amino Acid with Polar Interactions	Molecule	Affinity (Kcal/mol)	Amino Acid with Polar Interactions
GW0742	−11.1	His287 His413 Tyr437	GW0742	−8.5	Arg258
			GSK3787	−7.7	Trp228 Thr252
GSK3787	−9.1	Thr252 Asn307	GW0742	−8.1	Trp228 Lys229
			GSK3787	−7.4	Thr253 His413
L−165041	−8.7	His287 His413 Tyr437	L-165041	−8.3	Met192 Cys251 Thr252 Thr256 Ile290 Ala306
			GSK0660	−6.5	Arg198 Asn339
GSK0660	−8.6	Arg248 Thr252 Ala306 Asn307	L-165041	−8.1	Thr253 His413
			GSK0660	−8.9	Thr252 Thr253

**Table 3 ijms-22-03158-t003:** Treatments of tissues with two different combination types of agonists and antagonists: 1 μg/mL LPS; 100 nM GW0742; 10 μM L−165041; 10 μM GSK3782; 10 μM GSK0880; 10 μM 1400 W.

	Combination 1	Combination 2
Vehicle	0.01% DMSO	0.01% DMSO
LPS	LPS 1 μg/mL	LPS 1 μg/mL
LPS + agonist	LPS + GW0742	LPS + L-165041
LPS + antagonist	LPS + GSK3787	LPS + GSK0660
LPS + agonist + antagonist	LPS + GW0742 + GSK3787	LPS + L-165041 + GSK0660
LPS + 1400 W	LPS + 1400 W	LPS + 1400 W

## Data Availability

The data presented in this study are openly available in the repository: LM University of Brighton.

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
