# Peer review of "Co-Incubation with PPARβ/δ Agonists and Antagonists Modeled Using Computational Chemistry: Effect on LPS Induced Inflammatory Markers in Pulmonary Artery"

_ijms, 2021, doi:10.3390/ijms22063158_

Round 1

Reviewer 1 Report

This work was aimed at evaluating the involvement of PPARbeta/delta in the inflammation induced by LPS. To reach this objective, the authors used different combinations of agonists and antagonists, and rat pulmonary artery was employed as experimental model. Using computational chemistry approaches, the authors demonstrated that two ligands are able to simultaneously bind to PPARbeta/delta, and that the agonist binding followed by an antagonist administration switches PPARbeta/delta from transcriptional induction to transrepression. The manuscript provide interesting hints about the regulation of PPARbeta/delta by different modulators. However, there are some concerns that should be addressed in order to improve the rigor of the experimental design.

- The manuscript appears well written, however I recommend the authors to carefully check the whole text as some typographical errors are present.

- Paragraph 2.1: Results described in this section should be shown. In addition, the expression of PPARbeta/delta in pulmonary artery should be evaluated by immunohistochemistry in order to evaluate the relative expression in different cell types (endothelial or VSMC?)

- As also stated by the authors, the role of PPARbeta/delta in inflammation is complex, and opposite effects were often described in literature (also regarding specific gene targets), in dependence on the cell context and on the physiopathological condition. The authors should characterize the inflammatory markers in a broader manner. The analysis of IL6 and NO is too limited. Furthermore, since PPARbeta/delta are master regulators of gene transcription, these markers should be analysed not only by ELISA (or Western blot) but also by mRNA (or by luciferase assay).

- Data should be tested for normality distribution, especially in those cases where n=4-5. It is certainly possible that this small sample size does not follow a normal distribution. In this case, a non-parametric test should be employed instead of ANOVA. Otherwise, the sample size should be increased.

- Figure legends report that data are expressed as the mean ± S.E.M. The standard error of the mean indicates the uncertainty of how the sample mean represents the population mean. In my opinion, the authors inappropriately report the SEM instead of the Standard Deviation (SD). Since the SEM is always less than the SD, it deceives the reader into underestimating the variability between individuals within the study sample. Thus, SEM should be substituted by SD in all the sample descriptions.

- Procedures for tissue culture are incomplete. For instance, it is not mentioned whether serum was added in the culture medium. If yes, which concentration was used? In addition, cell death in tissue culture is a major concern. The authors should assess  cell viability of their tissue cultures in order to demonstrate the lack of necrosis/apoptosis.

Author Response

Thank you for the opportunity to revise our manuscript and resubmit. We have made the changes requested by the reviewers where possible. In light of the 7-day window to make changes and resubmit, it is not possible to make some of the changes requested by Reviewer 1. While many are very good points to make, these experiments were not conducted and thus the data is not available. Due to Covid-19 restrictions, it is not possible to conduct these experiments for the foreseeable future since the laboratories are restricted access. While it would be useful to do and would add to the overall picture of the project, they would not alter the underlying findings, meaning of the results and focus of this paper. We hope that the answers to the queries raised by the reviewers are sufficient and that the changes made in the manuscript address the insightful comments made by the reviewers. We feel that these changes have improve the manuscript and thank the reviewers for these points raised.

Reviewer 1

This work was aimed at evaluating the involvement of PPARβbeta/delta in the inflammation induced by LPS. To reach this objective, the authors used different combinations of agonists and antagonists, and rat pulmonary artery was employed as experimental model. Using computational chemistry approaches, the authors demonstrated that two ligands are able to simultaneously bind to PPARbeta/delta, and that the agonist binding followed by an antagonist administration switches PPARbeta/delta from transcriptional induction to transrepression. The manuscript provide interesting hints about the regulation of PPARbeta/delta by different modulators. However, there are some concerns that should be addressed in order to improve the rigor of the experimental design.

  1. The manuscript appears well written, however I recommend the authors to carefully check the whole text as some typographical errors are present.

We thank the reviewer for this opportunity to make corrections to the manuscript. A thorough review and rewording of the introduction and discussion has now been made, and we feel that this has much improved the focus of the paper.

  1. Paragraph 2.1: Results described in this section should be shown. In addition, the expression of PPARbeta/delta in pulmonary artery should be evaluated by immunohistochemistry in order to evaluate the relative expression in different cell types (endothelial or VSMC?)

The data outlined in paragraph 2.1 on the timed experiment has now been included in the supplementary section. These results show the changes in NO production over increased time of incubation, and explains the time point of 24 hours chosen for these experiments. The ELISA result is stated in the text, and it is not possible to conduct the experiment suggested by the reviewer. 

  1. As also stated by the authors, the role of PPARbeta/delta in inflammation is complex, and opposite effects were often described in literature (also regarding specific gene targets), in dependence on the cell context and on the physiopathological condition. The authors should characterize the inflammatory markers in a broader manner. The analysis of IL6 and NO is too limited. Furthermore, since PPARbeta/delta are master regulators of gene transcription, these markers should be analysed not only by ELISA (or Western blot) but also by mRNA (or by luciferase assay).

Thank you for this comment. Yes, we agree that PPARbd involvement in inflammation is complex, and is most notable in innate immune responses. We chose NO and Il6 as they are known to be controlled by PPARbd innate immune responses. The mRNA expression of iNOS is included in the manuscript, and the text has been clarified regarding these results.

  1. Data should be tested for normality distribution, especially in those cases where n=4-5. It is certainly possible that this small sample size does not follow a normal distribution. In this case, a non-parametric test should be employed instead of ANOVA. Otherwise, the sample size should be increased.

Upon re-analysis of data shown in Figure 2, it was found that the sets of data failed the normality test, and instead was analysed as suggested by Reviewer 1. This change has been included in the paper, and we thank the reviewer for this comment.

Reviewer 2 Report

The manuscript entitled “Co-incubation with PPARβ/δ agonists and antagonists; effect on LPS induced inflammatory markers in pulmonary artery” presents research on the possibility of binding two ligands simultaneously into the PPARβ/δ binding pocket. They suggest that co-incubation of agonists and antagonists to PPARβ/δ leads to a significant decrease in LPS-induced inflammation in rat pulmonary artery compared to single applications of each drug type.

The authors discuss the current achievements on the issues studied, referring to the relevant literature. They also describe a characteristic PPARβ/δ-ligand binding profile which is different for agonists and antagonists, and explain the mechanism of action by docking agonists and antagonists with PPARβ/δ, which confirms that agonist-antagonist co-binding may occur. The presented research is a multidirectional approach to PPARβ/δ research, which provides new information on its functioning at the molecular level.

Presented manuscript is interesting and correctly written, and in my opinion may be accepted for publication. A possible supplement would be to add a short last chapter entitled Conclusions.

Author Response

The manuscript entitled “Co-incubation with PPARβ/δ agonists and antagonists; effect on LPS induced inflammatory markers in pulmonary artery” presents research on the possibility of binding two ligands simultaneously into the PPARβ/δ binding pocket. They suggest that co-incubation of agonists and antagonists to PPARβ/δ leads to a significant decrease in LPS-induced inflammation in rat pulmonary artery compared to single applications of each drug type.

The authors discuss the current achievements on the issues studied, referring to the relevant literature. They also describe a characteristic PPARβ/δ-ligand binding profile which is different for agonists and antagonists, and explain the mechanism of action by docking agonists and antagonists with PPARβ/δ, which confirms that agonist-antagonist co-binding may occur. The presented research is a multidirectional approach to PPARβ/δ research, which provides new information on its functioning at the molecular level.

Presented manuscript is interesting and correctly written, and in my opinion may be accepted for publication.

  1. A possible supplement would be to add a short last chapter entitled Conclusions.

We thank reviewer 2 for their positive comments. We have included a short Conclusion as recommended. In light of this we also adjusted the title to reflect the computational chemistry docking focus of the manuscript.

Round 2

Reviewer 1 Report

The authors did their best to address the reviewer's comments. Even though they did not perform additional experiments because of COVID-19 pandemic restrictions, the manuscript is sufficiently improved.